# Understanding the Environment for Health-Promoting Schools Policies in Nova Scotia: A Comprehensive Scan at the Provincial and Regional School Level

**DOI:** 10.3390/ijerph18073411

**Published:** 2021-03-25

**Authors:** Anna Graham-DeMello, Joshua Yusuf, Margaret Kay-Arora, Camille L. Hancock Friesen, Sara F. L. Kirk

**Affiliations:** 1Healthy Populations Institute, Dalhousie University, 1318 Robie St, P.O. Box 15000, Halifax, NS B3H 4R2, Canada; js667453@dal.ca (J.Y.); M.Kay-Arora@dal.ca (M.K.-A.); Sara.Kirk@Dal.Ca (S.F.L.K.); 2School of Health and Human Performance, Dalhousie University, 1318 Robie St, P.O. Box 15000, Halifax, NS B3H 4R2, Canada; 3UT Southwestern Medical Center, Children’s Medical Center Dallas, 1935 Medical District Drive, Dallas, TX 75235, USA; Camille.HancockFriesen@UTSouthwestern.edu

**Keywords:** health-promoting schools, HPS, health policies, children’s health

## Abstract

The World Health Organization has identified the school community as a key setting for health promotion efforts, laying out its priorities in the Health-Promoting Schools (HPS) framework. This framework offers a comprehensive approach that has been adopted in countries around the globe, with defining characteristics focused around the school curriculum and environment. Nova Scotia (NS) adopted the HPS framework at a provincial level in 2005, but it has been variably implemented. We aimed to identify, categorize, and broadly describe the environment for HPS policies in NS. Four iterative steps were employed: (1) a scan of government and regional school websites to identify publicly available policies; (2) consultations with provincial departments with respect to policy relevance and scope; (3) cross-comparison of policies by two reviewers; (4) compilation of policies into an online database. Seventy policies at the provincial level and 509 policies across eight public school regions were identified. Policies focusing on a ‘safe school environment’ were most common; those addressing mental health and well-being, physical activity, nutrition and healthy eating, and substance use were among those least commonly identified. This scan provides a comprehensive overview of HPS-relevant policies in NS, along with relative proportions and growth over time. Our findings suggest areas of policy action and inaction that may help or hinder the implementation of HPS principles and values.

## 1. Introduction

Childhood and adolescence are known to be crucial periods that influence future health and well-being outcomes [1]. Health promotion and protection initiatives that are implemented early can help to reduce inequities and lay the foundation for a healthy and productive life [1]. There are ample opportunities, for instance, to prevent non-communicable diseases, promote good mental health, support academic outcomes and enhance resilience among populations through actions that focus on these early developmental periods [1]. This is important because children in Canada are not currently reaching their health potential. In a recent UNICEF report, Canada ranked 30th among 38 rich countries in terms of child and youth well-being [2]. Among 5–19-year-old children in Nova Scotia, only 39% are reaching at least 12,000 daily steps on average, a benchmark that is associated with health benefits [3]. This is slightly below the national average of 41%, a figure that has barely changed in over a decade [3]. Furthermore, 19.5% of Nova Scotian children live in food-insecure households; this proportion is marginally above the national average of 17.3% [4].

Health and education are provincial responsibilities, and healthy public policy is a core component of comprehensive approaches to health promotion. The World Health Organization recognizes the school community, in particular, as a key setting for health promotion efforts worldwide, laying out its priorities in the Health-Promoting Schools (HPS) framework, which was developed in the 1980s [5]. This framework offers a comprehensive and holistic approach that has been taken up in countries around the globe, with defining characteristics centering around school curriculum, school ethos and environment, and the wider community in which a school is situated [1].

A number of key implementation components of HPS have been identified, one of which centers around the development of supportive school policies [6]. There may be important learnings from enacting policies in various circumstances, and through tracking changes over time [7]. In addition, policy directories or databases can facilitate policy surveillance and monitoring; they may also facilitate communication and knowledge transfer between organizations, while helping to prevent unnecessary duplication of efforts among researchers and policy makers [7]. The Prevention Policies Directory, an online Canadian database of municipal, provincial/territorial, and federal policies aimed at chronic disease prevention, is one such example.

In Nova Scotia (NS), Canada, the provincial health and education systems have worked together to implement an HPS approach within the school system since 2005 [8]. A policy scan conducted in 2010 explored the context and nature of policies related to HPS across the province from 2003 to 2010 [8]. The review concluded that while supportive policies were present at different levels of jurisdiction, there was some incongruence in priorities and differences in enforcement practices between provincial-level and school region-level policies [8]. Furthermore, school regions were more likely to implement regulatory health promotion policies related to “safety” than other aspects of health promotion of relevance to the HPS model, such as school nutrition or physical activity policies [8]. Since this scan was conducted, restructuring of the provincial health and education systems has taken place, which may subsequently have impacted the policy environment in relation to the HPS approach. 

We aimed to identify, categorize, and broadly describe the policy environment in NS, in order to identify changes over time in the nature of policies that might support or hinder the adoption of the HPS framework. Methods employed include:Scanning both provincial government and school region websites to identify publicly available policies;Consulting with provincial departments with respect to policy relevance and scope;Cross-comparing policies to determine relevance and categorization;Compiling policies within an online database to facilitate data analysis.

## 2. Materials and Methods

For the purposes of this analysis, policies were broadly defined as statements and/or courses of action endorsed, implemented and resourced by the NS Provincial Government, or by at least one of eight public school regions. Provincial-level policies fall under the jurisdiction of various departments, including Nova Scotia Health (NSH), the Department of Health and Wellness (DHW), Education and Early Childhood Development (EECD), Communities, Culture and Heritage (CCH), the Department of Justice, and the Department of Transportation and Infrastructure Renewal. School region-level policies are those overseen by the seven English language Regional Centers for Education (RCEs) across the province, as well as one French first-language school board- the Conseil scolaire acadien provincial (CSAP). A variety of health promotion topics of relevance to the HPS framework were identified and defined in the early stages of this work; these were derived in part from a background paper from the International Union for Health Promotion and Education [9] in addition to other resources, including the World Health Organization’s School Environment Policies [10], the Ontario School Health Guidelines [11], and the Ontario Physical Health Education Association [12]. 

Our approach was guided by a previous NS-specific HPS policy scan [8] and consisted of four iterative steps: (1) identifying and defining health promotion topics of relevance to the HPS framework; (2) an online search to identify all publicly available policies of potential relevance in NS; (3) consulting with key stakeholders in the province to obtain feedback with respect to health promotion topics and their definitions; to check whether any policies had been overlooked, and to confirm that identified policies were still active and under the jurisdiction of a particular department; (4) an independent scan of all relevant policy documents by two reviewers (A.D. and J.Y.), to confirm relevance and categorization according to the pre-defined health promotion topic areas.

Across all NS provincial government departments, 115 policies were originally identified, and 766 were originally identified across all school regions. Policies were reviewed in sets of approximately 150 to assess agreement between two reviewers. To help ensure that all relevant policies were included, a traffic light system was adopted, which was applied while these reviewers independently scanned each policy document: the reviewers made note of health promotion topics of potential relevance, citing evidence from within each policy, and assigned a red, yellow, or green ‘light’ to each policy (red signaled exclusion; yellow signaled uncertainty and therefore need for further discussion, and green signaled inclusion). For each set of 150 policies, the two reviewers then met to compare their interpretations as to whether or not each policy should be included, and to reach consensus. The inter-coder agreement on the initial set of policies was approximately 65%. In response, the primary coder (A.D.) identified discrepancies and made additional modifications to health promotion topic definitions in order to improve outcomes. The inter-coder agreement on the second set of policies was approximately 76%. After minor additional changes to health promotion topic definitions, agreement across the final four sets of policies was approximately 88%. The reviewers were able to reach consensus across all 881 policies through further in-depth discussion and review of key policy information.

Once consensus across all policies was achieved, the reviewers extracted the data from each policy and summarized it within an online database according to relevant variables. These variables included: the name of the policy, reference dates, the jurisdiction under which it applied (i.e., provincial versus school region-level), the department responsible for overseeing it, the purpose of the policy, its target (e.g., school volunteers; students, school principals, etc.), and its assigned category (i.e., whether it was considered to be a Provincial Act, a formal policy, a set of written standards, a program, or a strategy) [13,14].

The HPS pillar under which each policy applied was also recorded within the database, as defined by the Pan-Canadian Joint Consortium for School Health (i.e., social and physical environment; teaching and learning; healthy school policy; partnerships and services) [15], along with the health promotion topic that each policy was found to align with (Table 1). Policies were included within the database if they were either directly or indirectly relevant to at least one HPS pillar and one health promotion topic. Only key health promotion topics identified within each policy (i.e., as informed by the preamble, purpose, intent, objectives, and/or similar upfront statements stated within the policy itself) were used to inform findings at both the provincial- and school region-level. 

*Airtable*, a freely available and open-source platform, was used for database development, allowing policy data to be housed online in an accessible format. Once populated, the database was exported to Microsoft Excel for further analysis. We produced a high-level description of policies that existed in NS as of January 2020 at both the provincial and school region level, according to both policy category and health promotion topic; we further described growth in the number of policies over time.

## 3. Results

Seventy policies at the provincial level and 509 policies across eight public school regions that were either directly or indirectly related to the HPS framework were identified.

### 3.1. Provincial-Level Policies

Across provincial government departments (including NSH, DHW, EECD, CCH, Justice, and Transportation and Infrastructure Renewal), all identified HPS-relevant policies were assigned one of five pre-defined policy categories. Findings revealed similar proportions of programs and strategies (3% and 7%, respectively), while Acts and formal policies were more frequent (20% and 29%, respectively). Written standards were most common, accounting for 41% of all identified provincial-level policy documents.

Policies developed by provincial government departments were examined according to the health promotion topic(s) they aligned with (Figure 1). Overall, findings revealed similar proportions of policies with a key focus on ‘transportation’ (2%), ‘transparency and accountability’ (2%), ‘mental health and well-being’ (3%), ‘community engagement’ (4%), ‘substance use’ (4%), ‘outdoor and indoor environment’ (5%), ‘healthy eating and nutrition’ (5%), and ‘personal development, relationships and sexual health’ (5%). Policies with a key focus on ‘physical activity’ and ‘youth-centered health services’ were more frequent (each at 7%). Policies focusing on ‘academic achievement and professional development’ (15%), ‘equity, inclusivity and accessibility’ (20%), and a ‘safe school environment’ (20%) were most common among all provincial-level policies. 

Growth in the number of provincial-level policies over time was further examined according to the initial policy implementation date (Figure 2). The overall number of policies increased from 35 to 81 (231%) from 2010 to 2020. Growth in ‘academic achievement and professional development’ policies aligned closely with the overall rate during this period, with the number of policies increasing from 6 to 14. Specific health promotion topics accrued new policies at higher rates during this period: the number of ‘safe school environment’ policies increased from 6 to 17; the number of ‘personal development, relationships, and sexual health’ policies increased from 1 to 3; the number of ‘outdoor and indoor environment’ policies increased from 1 to 4; and the number of ‘physical activity’-focused policies increased from 1 to 5.

Some health promotion topics accrued new policies at relatively lower rates during this period: the number of ‘equity, inclusivity and accessibility’ policies increased from 8 to 16; the number of ‘transportation’-focused policies increased from 1 to 2; the number of ‘youth-centered health services’ policies increased from 3 to 5; the number of ‘healthy eating and nutrition’ policies increased from 3 to 5, and the number of ‘mental health and well-being’ policies increased from 2 to 3. No new policies focusing on ‘transparency and accountability’ appeared to have been implemented during the 2010–2020 period. The first provincial-level policy with a key focus on ‘community engagement’ was adopted in 2014, while a second was adopted in 2015.

A separate analysis examining growth in the number of policies according to their date of latest revision (i.e., accounting for amendments that may have taken place over time) revealed that nearly three-quarters (73%) of all policies were either first implemented or had been revised over the past decade (since 2010). In considering specific health promotion topics, 60% of policies with a key focus on ‘youth-centered health services’ and ‘personal development, relationships and sexual health’ were either first implemented or have been revised since 2010; two-thirds (67%) of policies with a key focus on ‘outdoor and indoor environments’, ‘mental health and well-being’, and ‘equity, inclusivity and accessibility’ were either first implemented or had been revised in the same period. The vast majority of policies with a key focus on ‘academic achievement and professional development’ (79%), ‘physical activity’ (80%), and a ‘safe school environment’ (88%) were either first implemented or had been revised since 2010. Just one quarter of policies with a key focus on ‘healthy eating and nutrition’ were first implemented or revised since 2010. In considering extant provincial policies with a key focus on ‘transportation’, all were either first implemented or had been revised since 2011. Similarly, all policies with a key focus on ‘community engagement’ were either first implemented or revised since 2014, and all ‘substance use’-focused policies were either first implemented or had been revised since 2018.

### 3.2. School Region-Level Policies

Across all seven English school regions and the single French school region, all identified HPS-relevant policies were assigned one of five pre-defined policy ‘categories,’ as described in Table 1. Findings revealed that written standards accounted for 12% of identified documents, while formal policies were most common, accounting for 88% of all identified school region-level policy documents.

Policies developed across the school regions were also examined according to the health promotion topic(s) they align with (Figure 3). Overall, findings revealed similar proportions of policies with a key focus on ‘physical activity’ (0.3%), ‘mental health and well-being’ (1%), ‘healthy eating and nutrition’ (1%), ‘personal development, relationships and sexual health’ (2%), and ‘substance use’ (2%). Policies with a key focus on ‘outdoor and indoor environment’ and ‘youth-centered health services’ were slightly more common (each at 4%), while ‘transportation’-focused policies accounted for 8% of identified policies. More often, policies were found to focus on ‘community engagement’ (11%) ‘transparency and accountability’ (11%), ‘equity, inclusivity and accessibility’ (16%), and ‘academic achievement and professional development (16%). Policies focusing on a ‘safe school environment’ (24%) were most common among all school region-level policies.

The number of extant policies according to health promotion topic was further examined for each individual school region in the province (Figure 4). All eight regions had implemented at least one policy with a key focus on ‘outdoor and indoor environment’, ‘youth-centered health services’, ‘transportation’, ‘transparency and accountability’, ‘community engagement’, ‘academic achievement and professional development’, ‘equity, inclusivity and accessibility’, and a ‘safe school environment’. Policies with a key focus on ‘substance use’ had been implemented by seven of eight districts (88%); ‘personal development, relationships and sexual health’-focused policies by six of eight regions (75%); ‘healthy eating and nutrition’-focused policies by five of eight regions (63%), and ‘mental health and well-being’-focused policies by three of eight regions (37%). Only one of eight regions had implemented a policy with a key and upfront focus on physical activity.

Growth in the number of school region-level policies was also examined according to the initial policy implementation date (Figure 5), with findings revealing that the overall number of policies had increased from 374 to 532 (142%) between 2010 and 2020. For some health promotion topics, growth in policies during this period aligned closely with the overall rate: the number of ‘safe school environment’ policies increased from 91 to 128; the number of ‘transportation’ policies increased from 27 to 39; the number of ‘academic achievement and professional development’ policies increased from 61 to 88, and the number of ‘equity, inclusivity and accessibility’ policies increased from 54 to 82. Specific health promotion topics accrued new policies at relatively higher rates during this same period: the number of ‘youth-centered health services’ policies increased from 10 to 23; the number of ‘mental health and well-being’ policies increased from 2 to 5, and the number of ‘personal development, relationships, & sexual health’ policies increased from 3 to 8. Some health promotion topics accrued new policies at relatively lower rates during this period: the number of ‘healthy eating and nutrition’ policies increased from 5 to 6; the number of ‘community engagement’ policies increased from 50 to 59, and the number of ‘substance use’ policies increased from 11 to 12. No new policies with a key focus on ‘physical activity’ were implemented during the 2010–2020 period.

A separate analysis examining growth in the number of policies according to their date of latest revision (i.e., accounting for amendments that may have taken place over time) revealed that 57% of all policies were either first implemented or had been revised over the past decade (since 2010). In considering specific health promotion topics, the majority of policies with a key focus on the following topics were either first implemented or had been revised since 2010: ‘academic achievement and professional development’ (57%), ‘community engagement’ (58%); ‘equity, inclusivity and accessibility’ (61%); ‘safe school environment’ (66%); ‘transparency and accountability (69%); ‘outdoor and indoor environment’ (71%); ‘youth-centered health services’ (74%), ‘transportation’ (87%), and ‘personal development, relationships and sexual health’ (88%). Half of policies (50%) with a key focus on ‘healthy eating and nutrition’ were either first implemented or had been revised since 2010. No ‘physical activity’-focused policies had been implemented or revised since 2010. All school region-level policies with a key focus on ‘mental health and well-being’ were either first implemented or had been revised since 2012.

## 4. Discussion

Numerous policies have been developed in NS over recent years that are either directly or indirectly related to the HPS framework: 70 such policies were identified at the provincial level and 509 were identified across the eight public school regions in this scan. The nature and number of these policies offers insight into the provincial and broader policy environment of relevance to the implementation of HPS in this Canadian jurisdiction.

Across NS provincial government departments, the most common policies included those with a focus on ‘academic achievement and professional development’, ‘equity, inclusivity and accessibility’, and a ‘safe school environment’. The commonality of the latter, in particular, is not surprising, given the responsibilities of government and school systems to create safe spaces for students to learn, and is consistent with the previous NS-specific HPS policy scan [8]. More generally, findings appear to represent a shift away from more traditional HPS topics, such as physical activity, healthy eating, and sexual health, towards the inclusion of policies that could be viewed as either enablers or outcomes of HPS approaches, or that aim to create the conditions for improving health and well-being among students. Indeed, a recent systematic review identified policies as important ‘health assets’ that can serve to assist youth in advancing health and well-being for themselves and for others, citing the potential of a multi-level approach to health promotion in terms of supporting long-term behaviour change [16]. The notion of schools as health assets remains under-developed, however, and is worthy of further consideration given the key role that schools, and the policies enacted within them, can play in the lives of children and youth [16].

It is informative to parse provincial-level policies based on health promotion topic and growth rate, judged by numbers of policies, over time. Policies focusing on ‘healthy eating and nutrition’, for instance, make up just 5% of all identified provincial policies, and growth in this area has been relatively low over the past decade as compared with other HPS-relevant topic areas; furthermore, only one quarter of these policies were either first implemented or have been revised since 2010, which could signal that policy review efforts are indicated. This may also be timely given recent major changes to Canada’s Food Guide following an extensive nation-wide consultation process [17] and a national focus on physical activity through the Let’s Get Moving strategy [18].

Similarly, policies with a key focus on ‘mental health and well-being’ make up just 3% of all identified provincial policies, and growth in this area has been low relative to other topic areas over the past decade (the total number of policies increased from 2 to 3 in this period); two thirds of these policies were either first implemented or have been revised since 2010. While attention appears to have been given to this topic area within the past decade, it is important to consider whether or not this is sufficient, and if policy review efforts are presently indicated, given international attention to this issue [19]. In considering policies with a key focus on ‘physical activity’, although growth in this area has been relatively high as compared with others over the past decade (the total number of policies increased from 1 to 5), and the vast majority of policies were either first implemented or have been revised since 2010, this topic area is also fairly uncommon, making up just 7% of all identified provincial-level policies.

It is notable that the most recent UNICEF report focused on rising rates of childhood obesity as an indicator of poor health status, with healthy eating and physical activity being well recognized as two important behaviours for the prevention of obesity and non-communicable diseases [2]. With this in mind, greater attention to policies to promote healthy eating and physical activity is warranted given that the health status of school-aged children in Nova Scotia remains sub-optimal [20] and because of the role that schools can, and should, play in supporting the health, well-being and learning of children [5].

The number of policies developed by individual school regions varied considerably, both within and between health promotion topics. For instance, while ‘safe school environment’ policies are most numerous overall, there remains considerable variability between regions in terms of the number of policies implemented. Reasons for this could be myriad, and the number of policies on its own is not necessarily an indicator of comprehensiveness. Furthermore, policies falling under the heading of any one health promotion topic may vary in scope, target, and approach: ‘safe school environment’-focused policies may cover a range of issues, including, for example, personal privacy considerations, child abuse protections, and bullying. For these reasons, a focused review of individual health promotion topic areas would provide further context in terms of both policy comprehensiveness and coherence across regions.

In examining findings regarding school region-level policies, it is again useful to consider both the overall prevalence of particular health promotion topics and their growth rate over time. Policies focusing on ‘physical activity’ and ‘healthy eating and nutrition’, for instance, made up just 0.3% and 1%, respectively, of all identified school region-level policies, and growth in these areas has been low relative to other topic areas over the past decade; furthermore, just half of ‘healthy eating and nutrition’-focused policies were either first implemented or have been revised since 2010, and no ‘physical activity’-focused policies appear to have been implemented or revised during this period. This could signal a need to review and/or renew these subject areas, given the persistent and significant challenges posed by non-communicable diseases [21].

Policies with a key focus on ‘mental health and well-being’ made up just 1% of all identified school region policies, although growth in this topic area has been higher relative to others over the past decade [19]. In addition, all of these policies were either first implemented or have been revised since 2012. Again, while this topic area has gained greater attention in the past eight years or so, there remain few policies overall that are focused on the issue. It may be important for each region to consider whether or not this has been sufficient, and to determine if more in-depth policy review efforts are presently indicated.

Similar policy scans relating to HPS appear to be limited in the published international literature. A 2013 Australian review took a focused approach in identifying publicly available policies relating specifically to child sexual abuse prevention education in primary schools. Across both state and territory Departments of Education, the authors identified uneven system-level provision, and were able to develop a strategy and criteria to be applied in similar research internationally [22]. A 2018 Canadian review of equity-specific policies across 72 school boards in the province of Ontario revealed the under-representation of numerous topics, including religious accommodation, anti-racism and ethno-cultural discrimination, and anti-discrimination procedures for LGBTQ2+ students [23]. A study in Denmark that explored school health promotion policy implementation at national and regional levels noted that that health-promoting principles seemed to be ‘lost in translation’ from the national to the regional level, highlighting the complexity of policy implementation in the school setting [24].

An HPS policy scan specific to the province of Nova Scotia has previously been published [8]. A notable difference between the present scan and the one conducted previously is that nearly twice as many health promotion topics have now been included (twelve topics versus seven). The growth in health promotion topics likely relates to the expanded definitions and interpretations of HPS that have developed over the past decade, as identified through this analysis. While it is challenging to draw detailed comparisons, there are some notable similarities and differences between the two scans.

At the provincial level, both scans found that policies with a key focus on safety were most common overall (20% presently versus 25% in the earlier scan). While in the 2012 scan “safety” themed policies were followed closely in frequency by ‘mental health’-focused policies (20%), this proportion was considerably different in the present scan, with ‘mental health and well-being’-focused policies making up just 3% of identified provincial-level policies. Differences in the proportion of provincial policies with a key focus on the following health promotion topics are also noted: ‘physical activity’ (7% presently versus 13% previously); ‘healthy eating and nutrition’ (5% presently versus 12% previously); ‘substance use’ (4% presently versus 9% previously), and ‘personal development, relationships and sexual health’ (5% presently versus 8% previously).

At the school region level, both scans also found that policies with a key focus on safety were most common overall (24% presently versus 50% previously). While in the previous scan this was followed by policies with a focus on ‘mental health’ (17%), ‘physical activity’ (11%), ‘healthy eating’ (8%), and ‘substance use and misuse’ (7%), these proportions were considerably different in the present scan: ‘mental health and well-being’-focused policies made up just 1% of school region-level policies; ‘physical activity’ policies made up 0.3%, ‘healthy eating and nutrition’ policies made up 1%, and ‘substance use’ policies made up 2%. A focus on child safety is unsurprising, as is the lower proportion of policies that focus on health behaviours. Despite the HPS model being provincially endorsed in 2005, previous research has found that HPS is variably implemented in schools across the province, reflecting the challenges with focusing solely on educational outcomes rather than on more sustainable and integrated health and educational priorities in schools [8].

What is particularly striking is that the overall trajectory in terms of growth of HPS-relevant policies appears to be inverse to the trajectory of health outcomes we are witnessing in children and youth in Nova Scotia and elsewhere [25]. More is not necessarily better and the fact that intended health outcomes are lagging behind policy may represent the natural time-course of complex multifactorial health, education and social systems that cannot be expected to change quickly. It has also previously been noted that the introduction of new policies or practices in schools may displace the implementation of other policies or practices of proven benefit for students [21]. Given that the number of policies identified is not necessarily a reflection of importance, we contend that policy scope, reach and/or coherence may be better measures of impact.

This is the second scan to have been conducted that examines both provincial and school region-level policies that are supportive of the HPS framework in NS, Canada. This updated scan provides a comprehensive overview of the types of policies that exist across the province, along with relative proportions and growth over time according to HPS topic area. Our approach was comprehensive, involving close consultation with key stakeholders in the province so as to obtain feedback with respect to health promotion topics and their definitions, and to confirm that identified policies remained active.

There are several limitations worth noting with respect to this work. Health promotion topics and definitions used in this policy scan were developed by the authors, with input from key stakeholders. While each policy was then categorized according to the key health promotion topic(s) addressed within, this does not necessarily imply that other health promotion topics are not also represented, albeit to a lesser extent. A policy focusing on a school’s response to anaphylaxis among its students, for instance, might be assigned ‘safe school environment’, and ‘youth-centered health services’ as its key health promotion topics. If the procedural elements of the policy are fully considered, however, there may also be elements of ‘community engagement’ within this policy (e.g., there may be a provision that allows for community or parental engagement in the conception or design of allergy-safe spaces in schools). Our analysis has included only key health promotion topics, as identified by the authors. Another limitation of this work is that athough it provides some indication of where priorities lie, the number of identified policies cannot be relied upon as an indicator of policy comprehensiveness or coherence [21]. The same may be said for analyses of growth over time, although this does provide a sense of how priorities may be evolving over time. We recognize that there are likely to be a number of policies that are currently under review but that could not be included in our analysis, making it especially difficult to assess differences in terms of the province’s policy landscape between 2010 and 2020. This is in part due to a recent and significant shift within the education environment in NS, the outcome of which remains uncertain in terms of policy review and development. In addition, we acknowledge that further policy analyses based on school region may be of interest to some (e.g., in terms of differentiating between the types of policies developed in larger vs. smaller and urban vs. rural areas of the province). While this was outside the scope of this particular work, future research could address these aspects of policy development.

## 5. Conclusions

Leadership and support at regional and provincial/territorial levels is considered an essential condition for successful implementation of comprehensive school health approaches [26]. Policy development and implementation is a means to demonstrate such leadership, setting the tone for what matters to promote a safe and healthy environment for students [21]. This updated scan provides a comprehensive overview of HPS-relevant policies that exist across NS, along with relative proportions and growth over time according to health promotion topic. The incremental numbers and amendments of policies suggests that governmental agencies are responding to public concern about the health and well-being of children and youth. However, some health promotion topics appear to be under-represented within policies at both the provincial and school region level, with a focus remaining on policies that address safety in schools. Our findings highlight the complexity of HPS policy and the value of regular review and evaluation of such policies to ensure coherence and to support health and learning outcomes. Our approach provides a blueprint that other jurisdictions nationally and internationally can emulate.

## Figures and Tables

**Figure 1 ijerph-18-03411-f001:**
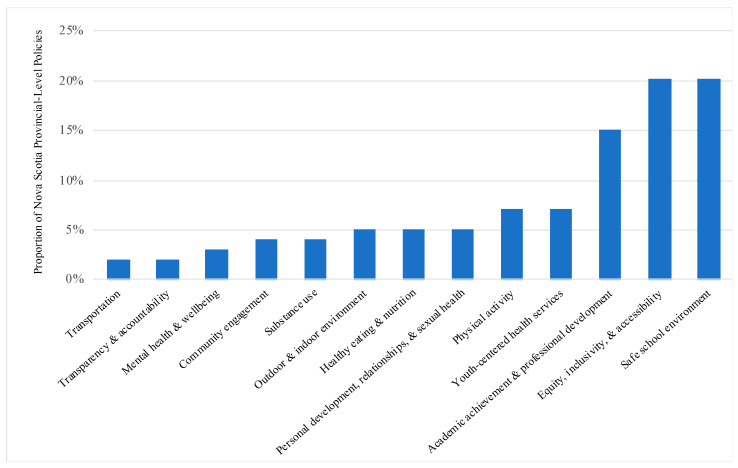
Proportion of Nova Scotia Provincial-Level Policies According to Health Promotion Topic Represented. Policies developed by Nova Scotia provincial government departments, including Education and Early Childhood Development; Nova Scotia Health; Communities, Culture and Heritage; Health and Wellness; Justice, and Transportation and Infrastructure Renewal, are proportionally represented according to the health promotion topic(s) they were found to align with.

**Figure 2 ijerph-18-03411-f002:**
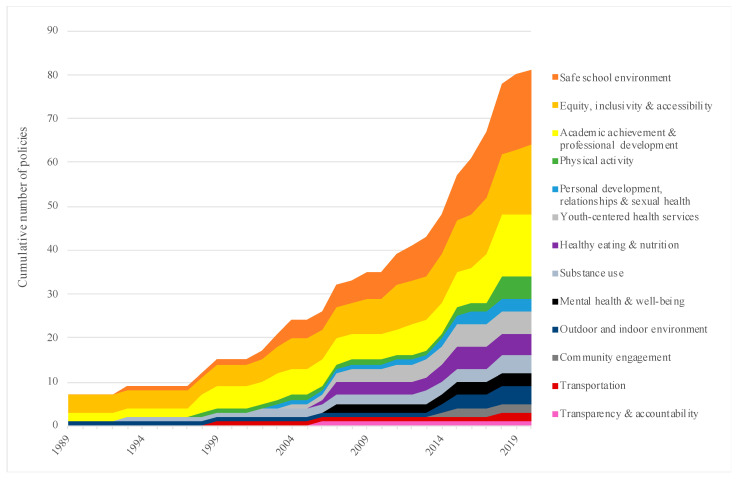
Growth in Nova Scotia Provincial-level Policies According to Health Promotion Topic: 1989–2020. Growth in the number of Nova Scotia provincial-level policies over time was examined according to the initial policy implementation date, allowing for improved understanding of which health promotion topics accrued new policies at higher or lower rates during this period.

**Figure 3 ijerph-18-03411-f003:**
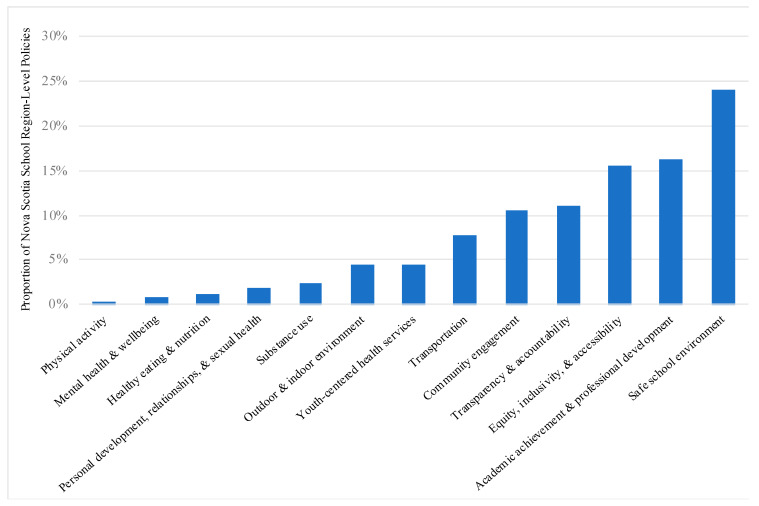
Proportion of Nova Scotia School Region-Level Policies According to Health Promotion Topic Represented. Policies developed across eight Nova Scotia school regions, including the French first-language Conseil Scolaire Acadien Provincial (CSAP), are proportionally represented according to the health promotion topic(s) they were found to align with.

**Figure 4 ijerph-18-03411-f004:**
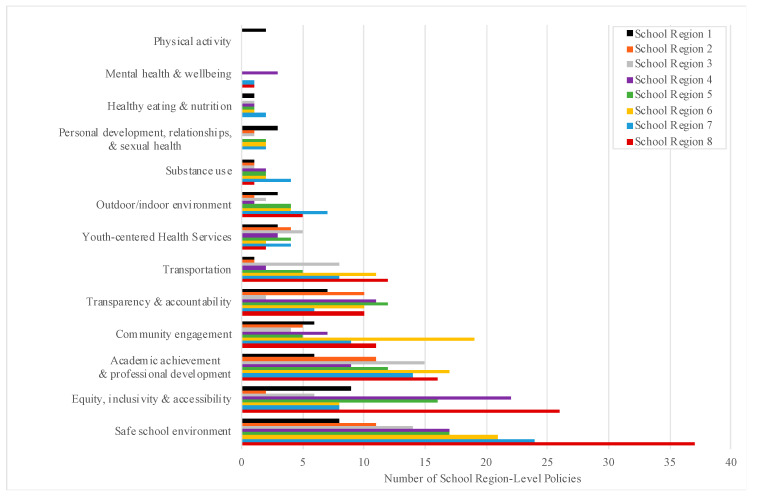
Number of Current Nova Scotia School Region-Level Policies According to Health Promotion Topic. The absolute number of existing policies corresponding to each health promotion topic was examined across eight individual school regions in the province.

**Figure 5 ijerph-18-03411-f005:**
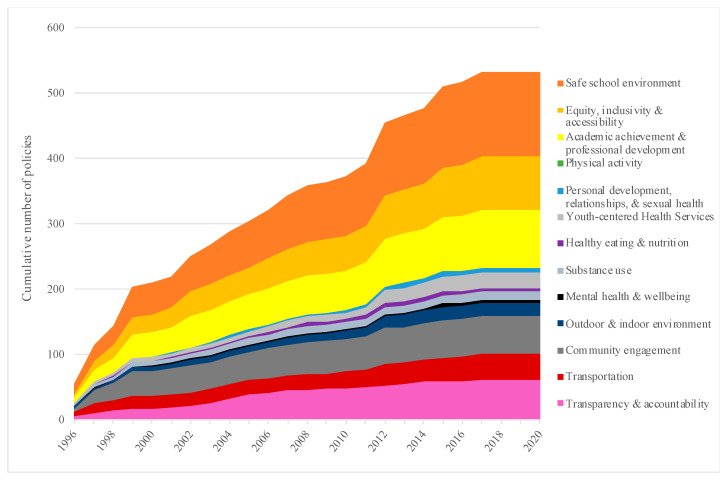
Growth in Nova Scotia School Region-level Policies According to Health Promotion Topic: 1996–2020. Growth in the number of Nova Scotia school region-level policies over time was examined according to the initial policy implementation date, allowing for improved understanding of which health promotion topics accrued new policies at higher or lower rates during this period.

**Table 1 ijerph-18-03411-t001:** Health Promotion topics and corresponding definitions of relevance to the Health-Promoting Schools Framework ^1^.

Topic	Definition
Healthy Eating & Nutrition	Policies that ensure students are not lacking proper nutrition; that facilitate affordable access to healthy food and drink; and that seek to enhance knowledge and skills in support of healthy food and beverage intake among students and others in the school community. Policies that develop food literacy through curricular, co-curricular and extracurricular activities. Policies that prevent marketing of, and sales and sales tactics surrounding, unhealthy foods.
Physical Activity	Policies that enable and encourage students and others in the school community to be more physically active and less sedentary during, before and/or after learning periods at school. Policies that develop physical literacy through curricular, co-curricular and extracurricular activities.
Transportation	Policies that support transportation options that enhance access to school, physical activity, and safety. Policies that encourage active transportation (e.g., walking, bicycling, skateboarding); discourage private vehicles; encourage public transit use; and expand school bus access.
Outdoor & Indoor Environment	Policies that support healthy outdoor and indoor school environments, and environmental health more generally. Areas of focus may include but are not limited to: accessibility (e.g., improving the physical environment); active outdoor learning; classroom arrangements; availability/quality of green space; school siting & design; building efficiency; air quality; waste management.
Substance Use	Policies relating to illicit and/or addictive substances that reduce use, address healthy equity, and support harm reduction and the development of life skills among students.
Personal Development, Relationships, & Sexual Health	Policies that support students’ growth, development, relationships and sexual health. Areas of focus may include but are not limited to: support for comprehensive sexual education curricula, supports for sexual minority and gender diverse students, supports for pregnant and parenting students, provision of menstrual products and contraceptives, supports aimed at fostering healthy relationships within school environments.
Mental Health & Well-being	Policies that seek to build the social, emotional and spiritual health and well-being of students and others. Areas of focus may include but are not limited to: coping and resilience; stigma and stereotypes; bullying, harassment, and discrimination; social-emotional learning; self-esteem; self-harm and suicide; mental health literacy; service provision standards; mental health emergency response.
Safe School Environment	Policies aimed at supporting safety, personal security, and privacy among students and other members of the school community, whether at school or during school-sanctioned events or activities. Areas of focus may include but are not limited to: violence, abuse, harassment, bullying; sexual misconduct; internet misuse and cyber protections; personal information and image-sharing; supervision of students; medical conditions and emergencies; disease and injury prevention.
Equity, Inclusivity & Accessibility	Policies that promote equity, inclusivity, and accessibility within the school community (e.g., via hiring practices, anti-discrimination and employee relations provisions; built environment and resource provision). Policies that promote equity and inclusivity with respect to certain population groups, including Indigenous peoples, African Nova Scotians, immigrants and refugees. Policies that support awareness, preservation, use, and celebration of knowledge, culture, and languages.
Community Engagement	Policies that support engagement with the wider community or region in which a school is situated. Areas of focus may include but are not limited to: student learning and socialization; research and extra-curricular opportunities; classroom activities involving community members; community partnerships for projects, use of space or resources; community involvement in school planning.
Transparency & Accountability	Policies aimed at promoting transparency and accountability within the school community. Areas of focus may include but are not limited to: staff performance; conflicts of interest; role modelling; ethical considerations; handling of complaints; disciplinary measures; fundraising requirements.
Academic Achievement & Professional Development	Policies that support students in achieving learning objectives, and staff members in advancing knowledge and skills. Areas of focus may include but are not limited to: student attendance; accelerated learning opportunities; provisions for staff training and knowledge enhancement.
Youth-Centered Health Services	Policies that generally support students’ physical and mental health via the provision of primary care services. Areas of focus may include but are not limited to: medical emergencies and conditions; allergy response; oral health; vaccinations; counselling.

^1^ Health Promotion topics and their definitions were derived in part from a background paper from the International Union for Health Promotion and Education [9] in addition to other resources, including the World Health Organization’s School Environment Policies [10], the Ontario School Health Guidelines [11], and the Ontario Physical Health Education Association [12].

## Data Availability

Data sharing not applicable. No new data were created or analyzed in this study. Data sharing is not applicable to this article.

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
