# Peer review of "Understanding the Environment for Health-Promoting Schools Policies in Nova Scotia: A Comprehensive Scan at the Provincial and Regional School Level"

_ijerph, 2021, doi:10.3390/ijerph18073411_

Round 1
Reviewer 1 Report
This is an excellent manuscript, and I thoroughly enjoyed reading it! I particularly liked how clear and concise the writing was, and could very easily follow along and articulate the methods and results sections.
The inter-rater reliability process was thoughtfully mapped out and presented to the reader. I have a few comments/questions for your consideration.
Did you notice any trends in school boards in terms of which types of policies they were developing/updating? Did anything stand out in terms of differentiating between larger and smaller boards, or rural/urban differences in policies? For example, did urban school boards have more policies (i.e., more students) than rural boards (i.e., fewer students)?
Overall, I believe this is a strong manuscript, and if consideration is given to the questions above, it is ready for publication. Thank you for the opportunity to review.
Reviewer 2 Report
The presented article addresses the interesting topic analyzing the policies that are at the core of the approach proposed by Health Promotion Schools and their presence in Nova Scotia. In my opinion the paper is well structured and clearly written. The description of the procedures by which the policy scan was prepared as well as other parts of the paper are well explained and detailed. In reviewing implemented policies, authors refer to a wide range of sources at both provincial and school region level.
In my opinion, there is a need for minor changes in the manuscript. The comments are basically only on the discussion part. Please consider supplementing the discussion with additional references. For example, in the section in lines 310-312 where the authors argue why the most popular policy "academic achievement and professional development" is, rightly noting that these can foster the implementation of the HPS approach and improve health and well-being. It is worth citing articles on the relationship of creating a motivational climate flowing from self-determination theory to improving student functioning or referring to developmental assets affecting health (for example: Van Bortel T, Wickramasinghe ND, Morgan A, et al. Health assets in a global context: a systematic review of the literature. BMJ Open 2019;9:e023810. doi:10.1136/bmjopen-2018-023810). I also suggest authors review the entire discussion in depth and try to supplement it with some additional references.
Summarizing, the paper is very interesting and contributes especially to Canadian health promotion by clearly presenting the list of existing policies meeting criteria of HPS framework and it should be accept for publication in IJERPH.
